

# Limited iron isotope variation among tissues of a marine fish: a case study of wild chub mackerel (*Scomber japonicus*)

Nanako Hasegawa*[1,2], Yoshio Takahashi[1], Takaaki Itai[1]

[1]Department of Earth and Planetary Science, The University of Tokyo, 7-3-1 Hongo, Bunkyo-ku, 8 Tokyo 113-0033, Japan
[2]Faculty of Veterinary Medicine, Hokkaido University, Kita 18, Nishi 9, Kita-ku, Sapporo 060-0818, Hokkaido Japan.

*Correspondence to*: Nanako Hasegawa (n-hasegawa@vetmed.hokudai.ac.jp)

**Abstract**

Iron homeostasis in marine organisms operates under chronically low iron bioavailability, which may shape the strategies of iron uptake and storage in fish. Stable iron isotope ratios ($\delta^{56}$Fe) have emerged as tracers of iron storage and
uptake in terrestrial mammals, yet the physiological drivers of isotope fractionation in marine organisms remain poorly understood. Here, we investigated $\delta^{56}$Fe variation and iron speciation across eight tissues of wild chub mackerel (*Scomber japonicus*), along with pool size estimation of key Fe species, including ferritin-bound (ferric) and heme-bound (mainly ferrous) Fe, using Fe K-edge X-ray Absorption Near Edge Structure (XANES) spectroscopy. The liver $\delta^{56}$Fe values were consistently higher than those of other tissues, with an apparent isotopic shift between ferritin- and heme-bound Fe ($\Delta^{56}$Fe) in
the liver averaging $2.04 \pm 0.22$‰ (2 S.D.). In contrast to the liver, no significant enrichment of heavy Fe isotope was observed in the ovary and red muscle despite their high ferritin-Fe contribution, suggesting a high interconversion rate between ferritin- and heme-bound Fe pools in these tissues. The overall range of $\delta^{56}$Fe variation among tissues was smaller than that reported in mammals. Our results indicated that muscular $\delta^{56}$Fe in marine teleost is primarily governed by source signatures and intestinal uptake efficiency, while tissue heterogeneity due to heavy Fe storage by ferritin exerts only a minor influence. These
findings highlight the potential of $\delta^{56}$Fe as a proxy for intestinal iron acquisition in fish and provide new geochemical perspectives on iron cycling through marine food webs.

**Short summary**

Iron stable isotope ratios provide a potential tracer of iron metabolism in fish. Here, we report tissue-specific isotope
variations in mackerel and evaluate how storage iron as ferritin affects fractionation using speciation analysis. The results show small isotopic differences among tissues, indicating that isotope ratios are primarily controlled by dietary values and intestinal uptake, highlighting the potential of natural isotope patterns as physiological indicators in fish.





## 1 Introduction

Iron is an essential element for life, functioning in electron transfer through the redox cycling between $Fe^{2+}$ and $Fe^{3+}$ (Georgiadis et al., 1992; Hoffbrand et al., 1976; Robbins and Pederson, 1970). While soluble $Fe^{2+}$ is highly bioavailable, insoluble $Fe^{3+}$ dominates under atmospheric conditions (Crichton and Boelaert, 2001), resulting in dissolved iron concentrations below $2\,nmol\,L^{-1}$ in the open ocean (Hatta et al., 2015; Johnson et al., 1997; Martin and Gordon, 1988; Nishioka and Obata, 2017). This scarcity constitutes a major factor limiting primary production by phytoplankton (Boyd et al., 2000; Martin et al., 1989; Martin and Gordon, 1988).

Although iron limitation in primary producers is well established, its implications for higher organisms such as fish remain less understood. Unlike phytoplankton, tissue iron concentrations in fish are not directly constrained by seawater iron availability (Galbraith et al., 2019). This is based on long-term iron retention strategies via ferritin (iron storage protein) and efficient intestinal absorption mechanisms specifically exist in marine fish (Bury et al., 2001; Carriquiriborde et al., 2004; Kwong and Niyogi, 2008). However, direct quantification of intestinal iron uptake has been largely restricted to laboratory animals using enriched isotopes, an approach that is not feasible for wild species especially marine fish.

Natural iron stable isotope ratios ($\delta^{56}Fe$) offer a broadly applicable, non-lethal proxy for assessing iron metabolism. $\delta^{56}Fe$ values are sensitive to intestinal absorption processes (Albalat et al., 2021; Flórez et al., 2017; Van Heghe et al., 2013; Hotz and Walczyk, 2013; Krayenbuehl et al., 2005; Stenberg et al., 2003) as well as to variation in iron storage and homeostasis, with ferritin-bound iron often implicated as a key driver of tissue-specific fractionation in mammals (Van Heghe et al., 2014; Hotz et al., 2012; Jaouen and Balter, 2014). Extending this framework to fish requires evaluating both intestinal absorption and ferritin-associated storage effects.

In this study, we examined $\delta^{56}Fe$ distributions across tissues of marine fish chub mackerel (*Scomber japonicus*). Additionally, X-ray Absorption Near-edge Structure (XANES) spectroscopy provided a non-destructive and oxidation state–sensitive approach to quantify ferritin-bound iron, allowing direct estimation of the proportion of $Fe^{3+}$ species in tissues without chemical extraction. Using this method, we quantified the relative contributions of ferritin- and heme-bound iron and assessed their roles in isotopic variability. By further considering physiological factors such as sex and reproductive status, our study provides new insights into the mechanisms governing $\delta^{56}Fe$ fractionation in fish and its potential as a tracer of intestinal iron acquisition in marine ecosystems.

## 2 Materials and Method

### 2.1 Sample Collection and Preparation

Six adult chub mackerel (females: Mk-1 to Mk-3, males: Mk-4 to Mk-6) were collected by longline fishing around Tsushima Island, East China Sea, on April 27, 2022. Fish were frozen onboard at −30 °C, stored at −20 to −25 °C for eight days, and subsequently transported to the laboratory at −15 °C. Body weight and fork length were recorded prior to dissection.



Total blood volume was estimated from teleost values (30–70 mL/kg; (Brill et al., 1998; Itazawa et al., 1983). Each individual was dissected into eight tissues (red and white muscle, liver, gonad, spleen, heart, gills, and blood). All tissue samples were stored at −80°C until the analysis. Wet weight was measured, and hepatosomatic index (HSI) and gonadosomatic index (GSI) were calculated as follows (Allaya et al., 2013; Shiraishi et al., 2008):

$$HSI = \frac{Liver\ weight}{Total\ body\ weight} \times 100 \tag{1}$$


$$GSI = \frac{Gonad\ weight}{Somatic\ weight} \times 100 \tag{2}$$

Tissues with high iron content (red muscle, liver, gonad, blood, and heart) were selected for Fe K-edge XANES analysis. Small frozen fragments were immediately sealed in oxygen-impermeable polycarbonate film (ASONE, Japan) after dissection and stored until measurement.

**2.2 Iron Concentration and Stable Isotope Measurement**

Analytical procedures followed (Hasegawa et al., 2023). Tissues were freeze-dried, and lipids removed from liver and gonad using chloroform:methanol mixture (2:1, v/v). Tests with certified reference materials confirmed no significant isotopic alteration during lipid removal (<0.07‰, n=4). Dried tissues (0.3–2.5 g) were digested in perfluoroalkoxy (PFA) vials using 68% $HNO_3$ and 35% $H_2O_2$ (both Tamapure AA-100, Tama Chemical Corp., Japan) at 120 °C. The digests were

evaporated to dryness and re-dissolved in 7M HCl containing 0.001% $H_2O_2$. Each sample was split for iron concentration analysis by inductively-coupled plasma mass spectroscopy (ICP-MS; Agilent 7700) and another for isotope analysis.

Iron purification was performed used anion-exchange chromatography following (Maréchal et al., 1999). Interfering elements (Ni, Cr, Cu) were removed with 7M HCl, and Fe was subsequently eluted with 2M HCl. The purified Fe fraction was dissolved in 2% $HNO_3$ at a concentration of 1 μg/mL, with Cu added for mass bias correction.

Stable isotope ratios were determined by multicollector-ICP-MS (Neptune Plus, ThermoFisher Scientific) at The University of Tokyo. Instrumental settings were: RF power 1200 W; Ar gas flow 15 L/min (cool), 0.7 L/min (auxiliary), 1.0 L/min (nebulizer). Signal intensity was 5–8 V for 1 μg/mL $^{56}Fe^+$, with blanks below 0.004 V. The $\delta^{56}Fe$ value was expressed relative to IRMM-014 as:

$$\delta^{56}Fe = \left[ \frac{(^{56}Fe/^{54}Fe)_{sample}}{(^{56}Fe/^{54}Fe)_{IRMM-014}} - 1 \right] \times 1000\ [‰] \tag{3}$$

Whole-body $\delta^{56}Fe$ was calculated as:

$$\delta^{56}Fe_{net} = \sum_i (\delta^{56}Fe_i \times T_i) = \sum_i (\delta^{56}Fe_i \times C_i \times M_i) \tag{4}$$

where $i$ represent the $i$-th tissue, and $T_i$, $C_i$, and $M_i$ represent the Fe burden, Fe concentration, and tissue mass, respectively.



Analytical accuracy was verified using certified reference materials (BCR-414, DORM-4, DOLT-5, ERM-CE464). Results were consistent with reported values: BCR-414 (−0.10 ± 0.03‰), DORM-4 (−0.24 ± 0.06‰), DOLT-5 (−2.32 ± 0.02‰),
ERM-CE464 (−0.55 ± 0.02‰).

### 2.3 XANES Spectroscopy

Iron K-edge XANES was analyzed at BL-12C, Photon Factory (KEK, Japan) following the protocol of Hasegawa et al. (2023). Frozen tissue fragments were analyzed in fluorescence mode using a Si detector, with energy calibration against
hematite (7110.2 eV). The number of independent spectral components was determined by principal component analysis (PCA).

The dominant Fe species in animal tissues include heme (Fe–porphyrin), ferritin (storage), and transferrin (Fe transport). Transferrin-bound Fe was excluded from further analysis due to minor abundance and negligible isotope fractionation (Walczyk and Von Blanckenburg, 2005). Thus, subsequent analyses focused on ferritin- and heme-bound Fe
species. Linear combination fitting (LCF) was performed to quantify their relative proportions in each tissue.

To account for possible post-mortem oxidation of hemoproteins, spectra of oxyHb, deoxyHb, and metHb were included in the fitting. Horse spleen ferritin (Hs-Ft) and bovine blood metHb (Bb-metHb) were used as reference standards, while oxyHb and deoxyHb were prepared from metHb following Di Iorio (1981), and Wilson et al. (2013). Pre-edge features were extracted and fitted using pseudo-Voigt functions implemented in Python. The quality of spectra fits was evaluated using
the R-factor, defined as:

$$R = \frac{\sum\{\chi_{obs}(E) - \chi_{cal}(E)\}^2}{\sum\{\chi_{obs}(E)\}^2} \tag{5}$$

### 2.4 Statistical Analysis

Statistical analyses were conducted in Python (ver. 3.9.19). Sex differences in physiological traits, Fe concentration,
and isotope ratios were tested using the Mann–Whitney U test.



## 3 Results

### 3.1 Physiological Parameters of Chub Mackerel

The six chub mackerel analyzed had fork lengths (mean ± 2 S.D.) of 37.7 ± 2.41 cm in females and 40.9 ± 4.00 cm
in males, and corresponding body weights of 0.71 ± 0.30 kg and 0.89 ± 0.24 kg, respectively (Table 1). Although females
tended to be slightly smaller, the difference in body length was not statistically significant (p = 0.06). Based on the growth
curve of (Shiraishi et al., 2008), all individuals were estimated to be at least four years old and sexually mature. The
hepatosomatic index (HSI) ranged from 0.74 to 2.14 in females, and the gonadosomatic index (GSI) ranged from 4.26 to 10.1
in males, consistent with spawning season in the East China Sea (Shiraishi et al., 2008). Among tissues, white muscle
represented the largest biomass fraction (31–52% of total body weight; Fig. S1), followed by gonads (3.8–8.9%), red muscle
(4.5–7.7%), gills (2.1–3.2%), liver (0.4–1.9%), heart (0.2–0.6%), and spleen (0.2–0.4%). No significant sex-related differences
were observed (p > 0.05). Red and white muscles, as well as gills, exhibited strong positive correlations with body weight (r
> 0.82, p < 0.04), whereas gonads showed a positive but non-significant trend (r = 0.77, p = 0.07; Fig. S2).

**Table 1:** Physiological parameters of chub mackerels.

|  | Mk-1 | Mk-2 | Mk-3 | Mk-4 | Mk-5 | Mk-6 | Mean |
|---|---|---|---|---|---|---|---|
| Sex | female | female | female | male | male | male |  |
| Body weight [kg] | 0.64 | 0.92 | 0.58 | 0.94 | 0.72 | 1 | 0.80 |
| Total length [cm] | 40.3 | 41.9 | 41 | 45.2 | 41.6 | 48.1 | 43.0 |
| Fork length [cm] | 37 | 39.4 | 36.7 | 40.8 | 38.5 | 43.4 | 39.3 |
| GSI | 10.1 | 10.1 | 4.26 | 8.76 | 10.0 | 7.25 | 8.41 |
| HSI | 2.14 | 1.48 | 1.05 | 1.07 | 0.74 | 1.06 | 1.26 |


### 3.2 Tissue Iron Concentrations and Stable Isotope Ratios

Mean iron concentrations (± 2 S.D.) were highest in the spleen (3,100 ± 970 µg/g), followed by blood (1,300 ± 530
µg/g), heart (720 ± 640 µg/g), liver (630 ± 560 µg/g), gills (290 ± 99 µg/g), red muscle (250 ± 92 µg/g), gonads (37 ± 26 µg/g),
and white muscle (15 ± 10 µg/g) (Fig. 1A). Male spleens contained significantly higher Fe concentrations than those of females
(p = 0.04), and liver and gonads also tended to be higher in males (p = 0.06).

Assuming a blood volume of 30 mL/kg, the total iron content in blood was estimated to be between 4.3 and 9.8 mg
(Fig. 1B). Red muscle contributed the next largest Fe pool (2.3–6.1 mg), followed by gills (1.3–2.0 mg), white muscle (0.9–
2.0 mg), spleen (0.8–2.6 mg), liver (0.5–1.6 mg), gonads (0.3–1.0 mg), and heart (0.2–0.9 mg). The total body Fe inventory
was therefore estimated to be approximately 28–59 mg/kg. Although the liver is the principal Fe storage tissue, it accounted
for only 2–5% of total body iron. In contrast, blood and red muscle together comprised 26–48% of the total iron burden.





Mean iron isotope compositions ($\delta^{56}$Fe ± 2 S.D.) of each tissue were as follows: red muscle, −1.54 ± 0.22‰; white muscle, −1.46 ± 0.23‰; liver, −1.19 ± 0.17‰; gonads, −1.23 ± 0.46‰; spleen, −1.37 ± 0.17‰; heart, −1.43 ± 0.21‰; gills, −1.37 ± 0.24‰; and blood, −1.39 ± 0.22‰ (Fig. 2). The mean $\delta^{56}$Fe across tissues was −1.40 ± 0.13‰, with the liver showing slightly higher values than other tissues. No significant sex differences were detected, although ovarian $\delta^{56}$Fe tended to be

lower than testicular values (p = 0.06).

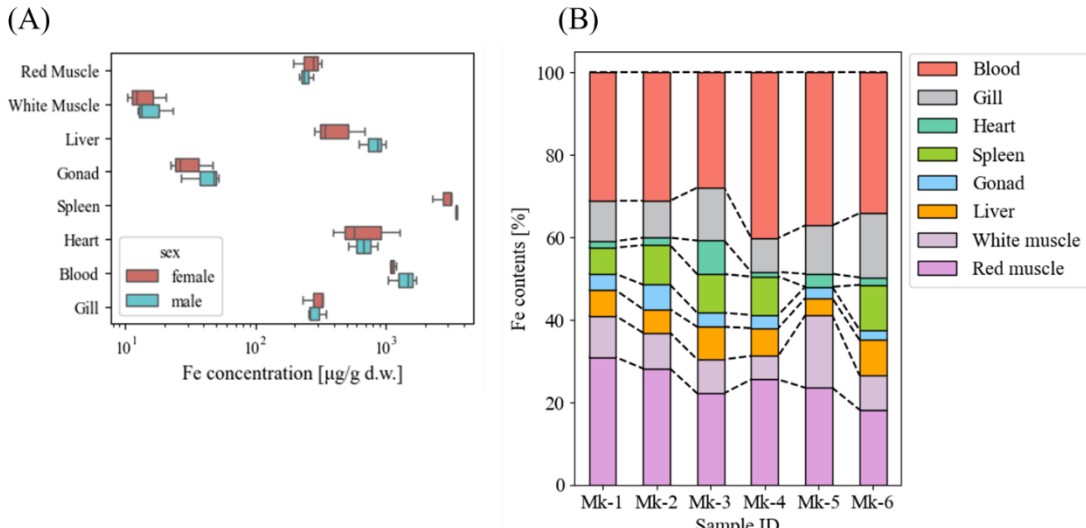

**Figure 1: (A) Iron concentration, and (B) Tissue iron burden of the chub mackerels (Mk-1 to Mk-3: female, Mk-4 to Mk-6: male)**




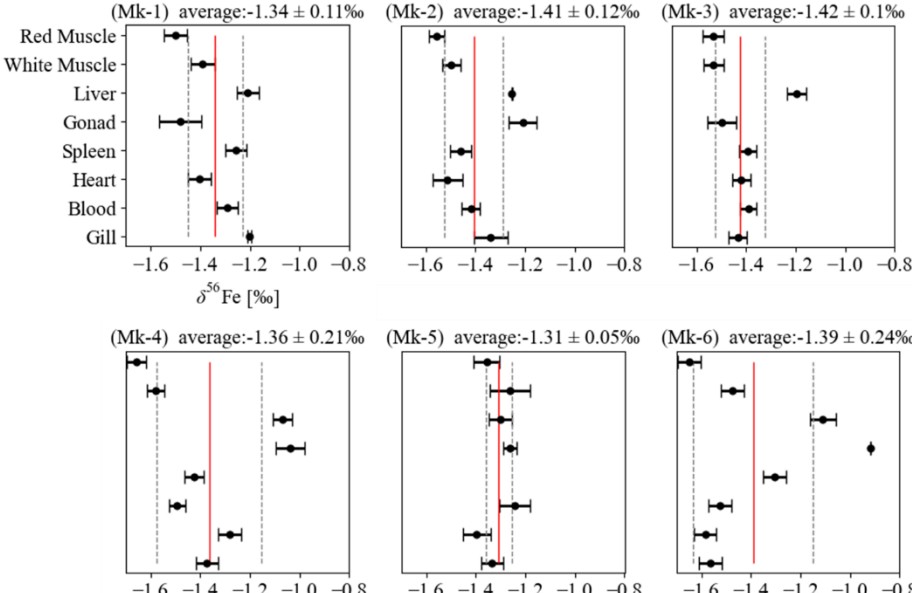

**Figure 2: Iron stable isotope ratio in tissues of the six *Scomber japonicus* individuals (Mk1-3: females, Mk4-6: males). The red line represents average $\delta^{56}$Fe values across the eight tissues. The gray dotted lines represent 2 S.D. values from the average line.**

## 3.3 Proportion of Ferritin-Bound Iron

Iron K-edge XANES spectra identified the dominant chemical forms of iron in each tissue (Wilke et al., 2001). Pre-edge peak positions for all samples were distributed between ferritin (most oxidized) and hemoglobin (reduced) reference standards (Fig. 3A).

Linear combination fitting (LCF) using Hs-Ft and Bb-Hb derivatives revealed that red muscle and heart were dominated by heme-bound Fe, whereas ferritin accounted for 17–29% of total Fe in the liver (Fig. 3B). Only one individual exhibited extremely low liver ferritin iron (8%). The liver ferritin fraction was comparable to those reported for chub mackerel from other regions (28%; Hasegawa et al., 2023) but substantially lower than those of mouse liver (66%; (Chen and Chen, 2018).

Sex-related differences were also apparent: females exhibited lower ferritin-bound Fe proportions in the liver, red muscle, and gonads than males ($p < 0.05$). In ovaries, ferritin represented the predominant Fe form (70–80%), whereas testes showed highly variable proportions (3–59%).



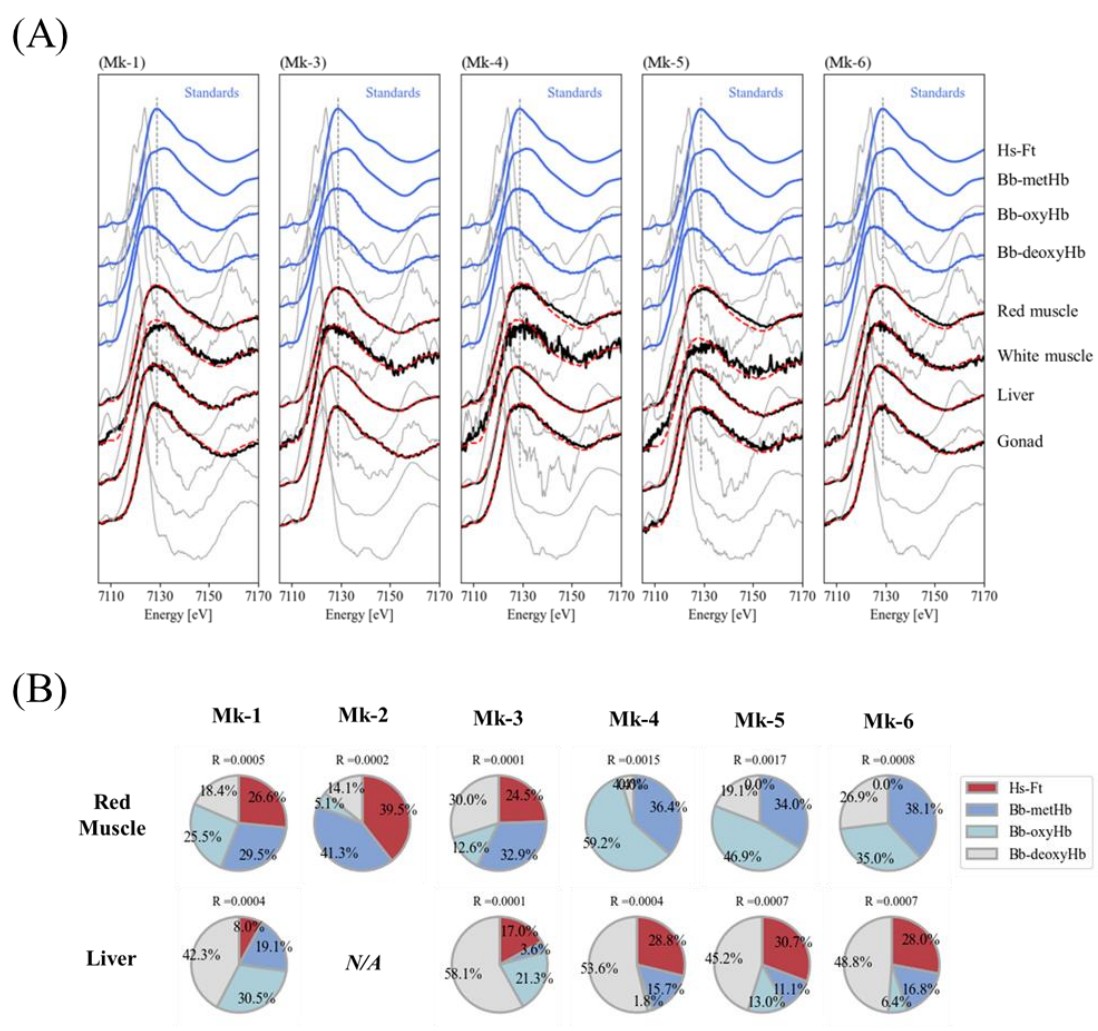

**Figure 3: (A) Fe K-edge XANES spectra of *Scomber japonicus* tissues (Mk-1 and Mk-3: female, Mk-4 to Mk-6: male). The blue lines represent the spectra of standard materials (Hs-Ft: horse spleen ferritin, Bb-metHb: bovine blood methemoglobin, Bb-oxyHb: bovine blood oxyhemoglobin, and Bb-deoxyHb: bovine blood deoxyhemoglobin). The gray lines represent 1st deviation of the row spectra. Dashed gray lines are the peak top energy of Hs-Ft (7129 eV). (B) Estimated proportions of each standard materials with linear combination fitting (LCF) of XANES spectra. R value is a goodness-of-fit parameter described in Eq. 4.**





## 4 Discussion

### 4.1 Elevated $\delta^{56}$Fe in Liver and Contribution of Ferritin-Bound Iron

The $\delta^{56}$Fe values in the liver of chub mackerel were consistently higher than those in other tissues, indicating an enrichment in the heavier Fe isotopes. Because ferritin preferentially incorporates in the heavier Fe isotopes during the oxidation in storage process (Albarède et al., 2011), this enrichment most likely reflects the contribution of ferritin-bound iron within the liver. However, the proportion of ferritin-bound Fe in mackerel liver (8–31%) was notably lower than that reported for mammalian liver (~60%; Chakrabarti et al., 2015; Chen and Chen, 2018), suggesting that a smaller fraction of hepatic Fe is stored as ferritin in teleost. Consequently, the $\delta^{56}$Fe enrichment in the mackerel liver appears more moderate than in mammals, possibly due to a greater influence of isotopically lighter Fe pools such as circulating hemoglobin iron.

Because blood contains little ferritin-bound iron, the elevated liver $\delta^{56}$Fe cannot be explained by residual blood contamination. Given that iron cycling within the body is a semi-closed system, in which the liver acts as the central exchange site between storage (ferritin) and transport (hemoglobin and transferrin) pools (Slusarczyk and Mleczko-Sanecka, 2021), hepatic $\delta^{56}$Fe can be interpreted as a mixture of these isotopically distinct components. This relationship can be expressed as:

$$\delta_{Liv} = p\delta_{Ft} + (1 - p)\,\delta_{Hm} \tag{6}$$

where p is the fraction of ferritin-bound Fe in the liver, and $\delta_{Ft}$ and $\delta_{Hm}$ represent the isotopic compositions of ferritin- and heme-bound Fe, respectively.

By assuming that the lowest $\delta^{56}$Fe observed in red muscle represents $\delta_{Hm}$, the isotopic offset between ferritin- and heme-bound Fe in the liver ($\Delta = \delta_{Ft} - \delta_{Hm}$) was estimated to be 1.99 ± 2.20‰ (2 S.D.). We note that this estimation shows relatively large variation because it includes one individual with nearly homogeneous $\delta^{56}$Fe across tissues (Mk-5) and another with an extremely low ferritin-bound Fe fraction (8%) in the liver (Mk-1). When these individuals were excluded, $\Delta$ was 2.04 ± 0.22‰. This value is slightly higher but comparable to previous observations in skipjack tuna (*Katsuwonus pelamis*, $\Delta$ = 1.52‰) and another chub mackerel population ($\Delta$ = 1.41‰; Hasegawa et al., 2023). Although additional data with a larger sample size are needed to obtain a more precise estimate of $\Delta$, the convergence of $\Delta$ among the remaining individuals supports the view that isotopic fractionation between ferritin and heme Fe represents a conserved biochemical signature.

### 4.2 $\delta^{56}$Fe Variation Across Tissues

Outside the liver, some tissues did not exhibit elevated $\delta^{56}$Fe values despite high ferritin content. In particular, ovaries and red muscle in females exhibited high ferritin-to-heme ratios but $\delta^{56}$Fe values comparable to other tissues. This pattern suggests that ferritin-bound Fe in these tissues is not retained over long periods but is actively mobilized and recycled in response to metabolic activity and oxygen consumption. For example, ferritin in skeletal muscle can be quickly mobilized to meet oxygen demand (Robach et al., 2007; Ryan et al., 2021), preventing the accumulation of a heavy isotope signature and maintaining $\delta^{56}$Fe values close to those of the whole-body iron pool dominated by heme iron. Moreover, $\delta^{56}$Fe has been



associated with oxidative stress markers including superoxide dismutase and glutathione peroxidase (Sauzéat et al., 2025), indicating that tissues prone to oxidative stress, such as gonads, may display isotopic variability due to redox-related iron turnover.

205       These observations suggest that ferritin-bound (storage) iron in chub mackerel exhibits a high turnover rate, resulting in small $\delta^{56}$Fe differences among tissues. In contrast, heme iron likely turns over more slowly. Fish erythrocyte lifespans range from ~80 to 500 days (Götting and Nikinmaa, 2017; Avery et al., 1992), comparable or longer than those of mice (~60 days, Dholakia et al., 2015) and humans (~120 days; Shemin and Rittenberg, 1946). Marine mammals, by contrast, display extended erythrocyte lifespans that scale with diving capacity (Pearson et al., 2024), highlighting the link between oxygen storage and 210  iron utilization. In migratory fish such as mackerel, the high and continuous demand for heme to support aerobic activity may promote rapid redistribution of heme iron, which may result in reduced $\delta^{56}$Fe variability among organs.

### 4.3 Features of $\delta^{56}$Fe Variation Among Marine Species

      The range of $\delta^{56}$Fe variation among tissues in chub mackerel was clearly narrower than that reported for mammals.
In skipjack tuna, muscular $\delta^{56}$Fe values are consistently high regardless of regional differences ($-1.46$‰ to $-0.71$‰; Hasegawa et al., 2022), whereas mammals generally show lower values ($-3.79$‰ to $-1.5$‰; Walczyk and von Blanckenburg, 2002; Balter et al., 2013). This pattern does not appear to arise from the enrichment of lighter isotopes in specific tissues, but rather from the limited influence of internal isotopic fractionation on whole-body $\delta^{56}$Fe. This interpretation is consistent with our previous study on a single individual (Hasegawa et al., 2023), and the limited inter-tissue $\delta^{56}$Fe variation observed across
multiple individuals in the present work supports that this characteristic is generally applicable to chub mackerel.

      As discussed in the previous section, hepatic ferritin iron likely undergoes isotopic fractionation on the order of approximately 1.5–2‰. However, because ferritin constitutes only a small fraction of total iron, its contribution to whole-body $\delta^{56}$Fe variation is limited. Consequently, heterogeneous distribution of $\delta^{56}$Fe across tissues is likely to have only a minor influence on the $\delta^{56}$Fe of indicator tissues representing whole-body $\delta^{56}$Fe values (e.g. muscle with a large Fe pool size). The
primary determinant of tissue $\delta^{56}$Fe in chub mackerel therefore appears to be iron uptake process from external sources rather than internal redistribution.

      The relatively high mean $\delta^{56}$Fe values in mackerel tissues may primarily reflect more efficient intestinal iron absorption than in mammals, resulting in a stronger reflection of environmental $\delta^{56}$Fe values. Additionally, selective ingestion of prey with intrinsically higher $\delta^{56}$Fe values could also contribute to their isotopic composition. Various prey taxa of marine
fish, including zooplankton, squid, and crustaceans, are known to exhibit relatively high $\delta^{56}$Fe values ($-1.00$‰ to$-0.03$‰; Von Blanckenburg et al., 2013; Hasegawa et al., 2022, 2023). Nevertheless, the $\delta^{56}$Fe differences observed among fish groups such as tuna and mackerel ($-1.58$‰ to$-0.71$‰) versus sardine and herring ($-2.64$‰ to$-1.73$‰; Hasegawa et al., 2022) are unlikely to be explained solely by isotopic differences in their prey, but are more likely attributed to variation in intestinal



iron absorption processes. However, recent studies have suggested species-specific $\delta^{56}$Fe variability even among zooplankton
(Hasegawa et al., submitted) and therefore prey-derived $\delta^{56}$Fe may also partially contribute to the overall isotopic composition of fish such as sardines and herrings that feed selectively on specific plankton groups. Further expansion of marine biological $\delta^{56}$Fe databases may enable a more comprehensive understanding of Fe physiology in wild marine organisms.

In contrast, terrestrial mammals such as humans, mice, and sheep consistently show large $\delta^{56}$Fe differences between liver and blood (Walczyk and von Blanckenburg, 2002; Balter et al., 2013), reflecting a well-developed iron recycling system
based on the long-term ferritin storage. This mechanism reduces daily iron requirements and promotes preferential uptake of lighter isotopes, thereby lowering the overall $\delta^{56}$Fe values. In contrast to mammals that rely on ferritin-based iron recycling, fish appear to lack a long-term iron storage strategy based on ferritin, despite inhabiting chronically iron-limited environments. They may instead have evolved an "absorption-dominant" mode of iron homeostasis that relies on efficient uptake of iron to sustain metabolic demands.


### 4.4 Implications for biological iron cycling in marine environments

The results of this study suggest that the $\delta^{56}$Fe variability in marine fish reflects dietary characteristics and uptake efficiency more strongly than internal physiological variations. The distinctly lighter isotopic signatures of animals compared to seawater may ultimately influence the isotopic composition of iron recycled through marine food webs. Therefore, whole-
body $\delta^{56}$Fe values could serve as an integrated tracer of biologically available iron within marine ecosystems. Our analyses also indicate that in fish, $\delta^{56}$Fe values in non-destructive tissues such as blood can represent the whole-body isotopic composition, highlighting their potential as key species for assessing iron supply and cycling in higher marine ecosystems.

In the future, applying a Rayleigh-type model may allow quantitative evaluation of the relative contributions of absorption and retention to isotopic fractionation in fish. Combined with XAFS analyses that provide species-level insights
into ferritin- and heme-bound iron pools, these approaches could offer a powerful framework for linking biological iron metabolism to marine biogeochemical iron cycling.

### 5 Conclusions

This study demonstrates that $\delta^{56}$Fe variations among tissues in chub mackerel are smaller than those observed in
mammals. Although isotopic fractionation of around 1.5-2‰ occurs in liver ferritin, its effect on whole-body $\delta^{56}$Fe is minimal due to the limited ferritin fraction and overall liver iron content. These results indicate that the high $\delta^{56}$Fe of mackerel primarily reflects dietary iron sources rather than internal isotopic fractionation. The relative homogeneity of $\delta^{56}$Fe among major tissues such as muscle and liver suggests that iron isotope pools in fish remain relatively stable, providing new insights into iron transport and isotope systematics within marine food web. Furthermore, the limited contribution of ferritin-bound iron and





small inter-tissue differences reflect characteristic iron absorption efficiency and internal distribution in mackerel, offering new perspectives for assessing physiological iron dynamics in fish. These findings provide fundamental information for studies of iron cycling and metabolism in marine organisms and have potential applications in food-web modeling and ecological assessments.




**Supplement**

This manuscript includes a supplementary material.

**Author contribution**

NH and TI contributed conceptualization, Methodology, and Project Administration; NH conducted Formal analysis, Software, Investigation, Visualization, and Writing (original draft preparation); YT and TI contributed Writing (review and editing); NH, YT and TI contributed Funding Acquisition.

**Data availability**

The data underlying this study are available in the article itself and its supplementary material.

**Competing interests**

The authors declare that they have no conflict of interest.

**Acknowledgement**

The authors appreciate Mr. Kei Zenimoto and Prof. Kotaro Shirai for collecting and transporting mackerel samples. We also thank Mr. Kotaro Hirayama and Mr. Yuma Sato for helping with the dissection of fish samples. We used ChatGPT to improve the English clarity and readability of the manuscript. This work was performed under the approval of the Photon Factory Program Advisory Committee (Proposal No. 2023G119, 2023S2-001, and 2022G126). This work was supported by JST
SPRING, Grant Number JPMJSP2108, and Asahi Group Foundation Grant Number A24-0057.





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
