# Peer review of "Limited iron isotope variation among tissues of a marine fish: a case study of wild chub mackerel (*Scomber japonicus*)"

_EGUsphere, 2025_

## Referee Comment (RC1)

Preprint by Hasegawa and colleagues report their work on six wild chub mackerel (*Scomber japonicus*) sampled in the East China Sea. They measured the Fe content and the isotopic ratio d56Fe in multiple tissues (liver, muscle, blood, gills, gonads, heart). The main result is small to none inter-tissue d56Fe variability within individuals, which the authors interpret as evidence that marine fish Fe isotopic composition reflects intestinal absorption and source composition. The methods seem sound and the results would make a nice addition to the literature on the topic.

However, I would recommend a major revision of the paper as there are discrepancies between the text, the data and the figures, and it should be revised carefully. In addition, the removal of 2 specimens out of a total of 6 with no valid reason other than getting a result closer to previous results (or maybe a reason not mentioned in the paper) should not serve as strong argument for the conclusion, nor the modified results be in the abstract, given the already small sample size.

Finally, measuring or extracting from other studies the signature of the potential sources (water and prey) would allow to make a more robust attribution and to demonstrate isotopic mixing or fractionation pathways.

**Major comments:**

1. My first main concern was the removal of two specimens (Mk-1 and Mk-5), after measurements were done, which produce a misleading narrow distribution. Either there is a strong incentive to remove these beforehand (e.g. sample compromised, biological justification), and it should be explicitly stated, or they should be included in the conclusion even if it adds more uncertainty. Moreover, the values resulting from the exclusion of these two specimens is mentioned as a key result in the abstract.

2. Secondly, I have noticed several discrepancies between the text and the values in the tables or the figures:
   - Maybe there is some factor that I don't know about that is missing, but when I take the wet weight in table S1 and multiply it by the Fe concentration (µmol/gww) in Table S2, lots of the results are different from the one given in the column "Total Fe (mg)" of table S2. It is especially true for blood Fe content (see tables at the end of this review). The small differences can be explained by approximations of both values but as for the others I couldn't figure out the cause behind the discrepancies. Maybe it has something to do with the conversion from wet weight to dry weight?
   - Page 5 line 131 "the total iron content in blood was estimated to be between 4.3 and 9.8 mg (Fig. 1B)" these values are incoherent with the ones given in Table S2.
   - Page 5 lines 132-133 again the ranges given are not the ones in Table S2 "gills (1.3-2.0 mg)" should be (1.3-3.7 mg), and "liver (0.5-1.6 mg)" should be (0.5-2.0 mg). Or did you remove sample Mk-5 on purpose here?
   - Page 7, l.155, section 3.3: Mk-1 and Mk-5 values do not seem to be included, otherwise the ranges given "17-29%" would be "8-31%" as it is the case on page 9 line 175.
   - Page 7 lines 159-160: it is stated that "females exhibit lower ferritin-bound Fe proportions in the liver, red muscle and gonads than males (p<0.05)". However, Figure 3B shows a higher proportion of ferritin (Hs-Ft, red portions) in red muscle and gonads in the female specimens. And in the following text the authors explicitly say that "ferritin represented the predominant Fe form (70-80%)" in ovaries "whereas testes showed highly variable proportions (3-59%)" in agreement with Figure 3B but in contradiction with the previous sentence.

3. Figure 1 (page 6) uses boxplots based on 3 values only. I am not convinced that this is the best statistical analysis one can do on such a small sized sample and especially when comparing groups (p.5 l128-130, section 3.2). Boxplots summarize a distribution and thus need more observations to be robust.

4. I would advise the authors to be more cautious with their conclusions as their sample size is small and present some variability.
5. The comparison between values seems subjective as the authors state "the d56Fe values in the liver [..] were consistently higher than those in other tissues", for a maximum difference of 0.35‰ between the liver and red muscle, and later say that the isotopic offset is "slightly higher but comparable to previous observation" for a difference of 0.46‰ at best. Either both are higher or both comparable in terms of differences. Or are there measurements for other organisms that show even larger isotopic offset? In that case I would understand the "comparable".
6. p.10 l231-236: it seems the two sentences contradict each other. The part stating "are more likely attributed to variation in intestinal..." excludes the contribution of prey d56Fe, which is not what you say before and after. Or can you demonstrate why the prey d56Fe cannot drive the variation in fish d56Fe?

**Other comments:**

- p2 l.39: "efficient intestinal absorption mechanisms **that** specifically exist in marine fish"
- p2 l.41: add references
- p3 l.77: "Iron purification was performed  using anion-exchange chromatography following Maréchal et al. (1999)."
- p4 l.103 "followind Di Iorio (1981) and Wilson"
- p5 l.117 "curve of Shiraishi et al. (2008), all…"
- p5 l127-129: precise "μg/gdw"
- p5 5 l.131 "the total iron content in blood was estimated to be between 4.3 and 9.8 mg (Fig. 1A)"
- p9 l.188: please check the value "1.99 +/-2.20 ‰". Using formula 6 and 1.66‰ as delta(hm) (lowest d56fe in red muscle here) I get a mean value of 2.7‰ with all samples and 2.24‰ excluding Mk-1 and Mk-5.
- p11 l.243-245: "efficient uptake of iron" can be misleading. Fish have low intestinal absorption of Fe, no excretion processes for Fe, which can be toxic, and as you show low Fe storage so it would appear that they limit absorption and intensively recycle absorbed Fe.

*Figures*:
- Fig. 2: Can you repeat the Y-axis legend at the far left of the male plots as well. It would make them easier to read.
- Fig. 3B: it would be easier to read the proportions were they around the discs instead of on them.
- Fig. S5: Point Mk-2 is missing on panel Liver "Total Fe" (0.97 in Table S2)
- Fig. S5: Y-axis seems wrong. Gonads "Total Fe" in mg is lower than 1 in Table S2 but you show data going up to 20 on the y-axis.

- Bibliography: "Von Blackenburg" and "Von Heghe" should be at Vs, "Di Iorio" at Ds.

|  |  | g ww | [Fe] µg/gww | Fe (mg) | In Table S2 |
|---|---|---|---|---|---|
| Mk-1 | red muscle | 31,1 | 144,36 | 4,49 | 4,12 |
|  | white muscle | 333 | 4,01 | 1,34 | 1,33 |
|  | liver | 13,7 | 68,95 | 0,94 | 0,83 |
|  | gonad | 58,6 | 9,34 | 0,55 | 0,53 |
|  | spleen | 2,7 | 720,77 | 1,95 | 0,85 |
|  | heart | 4 | 120,72 | 0,48 | 0,21 |
|  | blood | 5,1 | 222,27 | 1,13 | 8,29 |
|  | gill | 14,7 | 96,15 | 1,41 | 1,3 |
|  |  |  |  |  |  |
| Mk-2 | red muscle | 52,7 | 93,45 | 4,92 | 4,77 |
|  | white muscle | 326 | 4,5 | 1,47 | 1,47 |
|  | liver | 13,6 | 80,49 | 1,09 | 0,97 |
|  | gonad | 84,3 | 12,32 | 1,04 | 1 |
|  | spleen | 4,8 | 431 | 2,07 | 1,62 |
|  | heart | 5,6 | 70,43 | 0,39 | 0,3 |
|  | blood | 4,6 | 196,19 | 0,90 | 10,51 |
|  | gill | 25,6 | 63,07 | 1,61 | 1,51 |
|  |  |  |  |  |  |
| Mk-3 | red muscle | 36,5 | 68,94 | 2,52 | 2,34 |
|  | white muscle | 182 | 4,77 | 0,87 | 0,87 |
|  | liver | 6,1 | 181,92 | 1,11 | 0,83 |
|  | gonad | 23,7 | 16,08 | 0,38 | 0,36 |
|  | spleen | 2,8 | 891,42 | 2,50 | 0,99 |
|  | heart | 5,1 | 262,33 | 1,34 | 0,86 |
|  | blood | 1,8 | 174,24 | 0,31 | 5,89 |
|  | gill | na | 73,61 | na | 1,35 |
|  |  |  |  |  |  |
| Mk-4 | red muscle | 73,8 | 84,66 | 6,25 | 6,09 |
|  | white muscle | 374 | 3,64 | 1,36 | 1,36 |
|  | liver | 10,1 | 234,24 | 2,37 | 1,61 |
|  | gonad | 75,7 | 10,68 | 0,81 | 0,76 |
|  | spleen | 5 | 872,5 | 4,36 | 2,22 |
|  | heart | 4,6 | 120,75 | 0,56 | 0,27 |
|  | blood | 5,5 | 349,12 | 1,92 | 19,12 |
|  | gill | 31 | 68,96 | 2,14 | 1,97 |
|  |  |  |  |  |  |
| Mk-5 | red muscle | 39,1 | 74,99 | 2,93 | 2,75 |
|  | white muscle | 326 | 6,28 | 2,05 | 2,05 |
|  | liver | 5,3 | 161,98 | 0,86 | 0,49 |
|  | gonad | 65,7 | 5,29 | 0,35 | 0,32 |
|  | spleen | na | na | na | na |
|  | heart | 5,6 | 144,24 | 0,81 | 0,37 |
|  | blood | 6,1 | 206,17 | 1,26 | 8,65 |
|  | gill | 21,1 | 65,12 | 1,37 | 1,37 |
|  |  |  |  |  |  |
| Mk-6 | red muscle | 57,7 | 77,27 | 4,46 | 4,28 |
|  | white muscle | 513 | 3,85 | 1,98 | 1,98 |
|  | liver | 10,6 | 259,06 | 2,75 | 2,01 |
|  | gonad | 67,5 | 9,17 | 0,62 | 0,56 |
|  | spleen | 5,2 | 875,43 | 4,55 | 2,56 |
|  | heart | 3,5 | 185,29 | 0,65 | 0,42 |
|  | blood | 12,5 | 275 | 3,44 | 16,02 |
|  | gill | 40,9 | 94,75 | 3,88 | 3,65 |